# Optimization of the Outlet Shape of an Air Circulation System for Reduction of Indoor Temperature Difference

**DOI:** 10.3390/s23052570

**Published:** 2023-02-25

**Authors:** Jin-Young Park, Young-Jun Yoo, Young-Choon Kim

**Affiliations:** 1Division of Mechanical Engineering, Kongju National University, Cheonan-si 31080, Republic of Korea; 2Doohyun Engineering & Construction Co., Ltd., Hongseong-gun 32219, Republic of Korea; 3Division of Mechanical and Automotive Engineering, Kongju National University, Cheonan-si 31080, Republic of Korea

**Keywords:** air circulator, smart farm, flow analysis, temperature anomaly

## Abstract

This study proposes an air circulation system that can forcibly circulate the lowest cold air to the top of indoor smart farms, and it has a width, length, and height of 6, 12, and 2.5 m, respectively, to reduce the effect of temperature differences between the upper and lower parts on the growth rate of plants in winter. This study also aimed to reduce the temperature deviation generated between the upper and lower parts of the target indoor space by optimizing the shape of the manufactured outlet of the air circulation system. A table of L9 orthogonal arrays, which is a design of experiment methodology, was used, and it presented three levels of the following design variables: blade angle, blade number, output height, and flow radius. Flow analysis was performed for the experiments on the nine models to minimize the high time and cost requirements. Based on the derived analysis results, an optimized prototype was manufactured by applying the Taguchi method, and experiments were conducted by installing 54 temperature points in an indoor space to identify the temperature difference between the upper and lower parts over time for the performance experiment. Under natural convection, the minimum temperature deviation was 2.2 °C and the temperature difference between the upper and lower parts did not decrease. For a model without an outlet shape, such as a vertical fan, the minimum temperature deviation was 0.8 °C and at least 530 s were required to reach a difference of less than 2 °C. When air was circulated in the air circulation system with the proposed outlet shape, the minimum temperature deviation was 0.6 °C and the time required to reach a difference of less than 2 °C was 440 s. Using the proposed air circulation system, cooling and heating costs are expected to be reduced in summer and winter because the arrival time and temperature difference between the upper and lower parts can be reduced using the outlet shape compared with the case without the outlet shape.

## 1. Introduction

As urbanization increases, younger generations are becoming concentrated in metropolitan areas, the influx of younger generations in rural areas is decreasing, and the young workforce in rural areas continues to decrease. Moreover, a decrease in crop production associated with climate change and an increase in labor costs resulting from industrial development are threatening the livelihoods of farmers. As such, the number of local farms has been gradually decreasing [1,2,3,4,5].

A smart farm employs control measurement technologies and information and communications technologies (ICT) indoors through various sensors, and it optimizes the environment, such as the temperature and humidity, to suit the growth characteristics of plants. Research and development on smart farms has been actively conducted because these farms can reduce deviations in crop yields. Lee researched and analyzed areas that need to be developed and improved regarding excessively expensive smart-farm technology [1]. To examine the effect of ICT on labor reduction and productivity improvements when cultivating cherry tomatoes, Kim conducted a study to investigate and compare ICT applications, cherry tomato growth and quantity in general farmhouses, and working hours spent on environmental management and nutrient solution management in single-span greenhouses with poor facilities, which represent the majority of cherry tomato cultivation systems [2]. Culibrina et al. studied the control of farm power distribution and irrigation system; this paper proposes a communication methodology of the wireless sensor network for collecting environment data and sending control commands to turn the irrigation system on/off and manipulate power distribution [6]. According to data released by the Ministry of Agriculture, Food, and Rural Affairs of South Korea, the number of smart farms increased by 2368 units in 2020 compared to 2017, and the area further increased by 1938 ha [7]. Since the smart farm is included in the farming area, the farming area increases and the smart farm are organically connected. In addition, many people are interested in agriculture because of the lower entry barriers of smart farms after the 4th industrial revolution. Moreover, production has increased by 32.1% and working hours and pests have decreased by 13.8% and 6.2%, respectively [8]. Furthermore, 59.5% of current farmers are willing to introduce smart farms in the future [9].

Because plants are significantly affected by temperature and humidity, these parameters are controlled inside the smart farm. However, even inside a smart farm, the growth rate of crops varies because of the temperature difference between the upper and lower parts of the structure [10,11,12,13]. To address this challenge, research has been conducted on methods of forcibly circulating air horizontally and vertically in the upper part. Yu et al. conducted a study on the effect of horizontal airflow created by circulating fans on the horizontal and vertical distributions of environmental factors, including temperature, humidity, CO_2_ concentration, and wind speed [3]. To minimize the indoor temperature difference, Kang et al. investigated and analyzed the temperature inside a greenhouse that used a vertical air circulation fan. However, the vertical and horizontal air circulation methods do not generate smooth air circulation, and only partial air circulation occurs [14]. Kuroyanagi et al. discussed the necessity of air circulation devices and the problems of tunnel-type air circulation devices in relation to the yield, disease control, and circulator performance required by indoor farms [15]. Murakami et al. studied the smart farming system in a limited, enclosed area wherein different sensors were strategically positioned to measure parameters such as moisture content, temperature, pressure, light intensity, and pH of the soil [16].

To improve partial air circulation, studies have been conducted on air circulation methods associated with ventilation devices or structures [17,18,19,20]. Jhin Ho developed a “heat storage roof with an air circulation structure” to utilize the roof of an existing concrete structure as a solar system, which is commonly observed in South Korea, analyzed its heat transfer characteristics and thermal performance through experiments and theoretical analysis, and proposed associated designs [5]. Cheboxarov and Vik et al. developed a model of a large-scale cross-flow wind turbine with rotating blades, which was presented, and conducted numerical and aerodynamic studies of the turbine [21]. Lim et al. conducted a study to confirm the temperature and humidity change and distribution uniformity in a high-temperature glass greenhouse under three treatments: air flow fan, fog system, and both air flow fan and fog system [22].

Lee et al. installed a counterflow ventilation system in a tomato cultivation multi-span greenhouse located in Bongdong-eup, Wanju-gun, Jeollabuk-do to control humidity in the greenhouse in winter, and they operated it at night in winter to analyze the effect of controlling humidity in the greenhouse and improving the crop growth environment [23]. Koh et al. studied the effect of changes in the location and speed of the air exhaust port to investigate the ventilation effect on the indoor contaminated air of an air-exhaust-integrated air purifier installed on the wall [24]. Murakami et al. conducted a simulation-based ventilation characteristic analysis to apply a new scale to the indoor ventilation effect [16].

Temperature is one of the most sensitive factors for plant growth; thus, many researchers have studied the effects of air circulation on temperature. However, existing studies have primarily investigated the effects of installing fans horizontally and vertically or increasing the number of fans. Such studies are limited because the air circulation at the top and bottom is not smooth; thus, further studies are required to uniformly maintain the indoor temperature distribution by circulating the lowermost air to the top through the air flow path. Thus, in this study, we proposed an air circulation system that can forcibly circulate cold air from the bottom to the top in indoor space smart farms with widths, lengths, and heights of 6, 12, and 2.5 m, respectively. The objective of this study was to reduce the temperature difference between the upper and lower parts of the target indoor space by optimizing the shape of the manufactured outlet of the air circulation system.

## 2. Theoretical Background

### 2.1. Design of Experiment

Design of experiment is a method of determining a plan for conducting experiments and analyzing experimental data, and it can be regarded as a method for planning experiments to solve problems, collect data, and analyze data, with the objective of obtaining a great deal of information in the minimal number of experiments [13,14,18].

In general, after conducting an experiment, the shape of the data cannot clearly indicate, either theoretically or empirically, how the characteristic value affects the experimental variable or relates to the results because of the many factors that affect the characteristics. For example, the distributions of differences in raw materials, experimental devices, and individual abilities are affected by environmental changes and sample errors. Therefore, prior to conducting the experiment, the design of experiment approach was used to reduce the trial and error involved in achieving the desired purpose by planning the experiments.

#### 2.1.1. Taguchi Method

The Taguchi method is one of the design of experiment methodologies, and it is an effective method of obtaining a great deal of information based on a small number of experiments. Optimal conditions are obtained by identifying factors that significantly affect the experimental results using a table of orthogonal arrays and the signal-to-noise ratio (S/N ratio).

#### 2.1.2. Loss Function

A loss function is used to reasonably evaluate quality. The y variable in the loss function is the characteristic value of the product that can be converted into a value amount. If the target value of the characteristic value y is m, then the economic loss that occurs when it deviates from the target value m is defined as L(y) [14,25,26].
(1)$$L(y)=k{\left(\frac{1}{y}\right)}^{2}=A_{0}\Delta^{2}\left(\frac{1}{y^{2}}\right)$$
where *k* represents the constant of proportionality, *A_0_* represents the loss when not functioning, and Δ represents the functional limit.

The loss function can be classified into nominal-the-best, smaller-the-better, and larger-the-better characteristics depending on the type of target value. In this study, the airflow speed of the indoor air circulation system was selected as a characteristic and larger-the-better characteristics were used because a faster flow rate led to better results.

Larger-the-better applies to a characteristic that is better if the value of the characteristic y is larger but not negative. This characteristic is opposite to the smaller-the-better characteristic. The loss function can be expressed as Equation (1), and the loss function graph is shown in Figure 1.

### 2.2. Analysis Model

A solution for a series of nonlinear partial differential equations is required for the analysis of indoor flow using computational fluid dynamics. The partial differential equation is a flow field governing equation, such as the law of conservation of mass and momentum and the energy equation. To solve the governing equation, discretization and linearization must be performed. During this process, algebraic equations are repeatedly calculated to predict the distribution of the pressure and velocity at the point where discretization is performed.

#### 2.2.1. Flow Field Governing Equation

(1)Law of Conservation of Mass

The law of conservation of mass states that an increase in mass within a control volume is equal to the magnitude of the mass entering through the control volume. Therefore, an increase in mass can be expressed as shown in Equation (2) [27,28]. However, it is assumed that there is no flow owing to thermal energy because it takes a very long time and high-specification workstations to consider both natural convection flow and forced convection flow by air circulation devices.
(2)$$\frac{\partial \rho }{\partial t}+\Delta \left(\rho u_{i}\right)=0$$

(2)Momentum Equation

Newton’s second law states that the sum of forces acting on the control volume equals the change in the momentum of the control volume, and it is expressed as Equation (3).
(3)$$\frac{\partial }{\partial t}\left(\rho u_{i}\right)+\frac{\partial }{\partial x_{i}}\left(\rho u_{i}u_{j}\right)=-\frac{\partial \rho }{\partial x_{i}}+\frac{\partial }{\partial x_{j}}\left\{\mu \left(\frac{\partial u_{i}}{\partial x_{j}}\right)+\left(\frac{\partial u_{j}}{\partial x_{i}}\right)\right\}+\rho g_{i}$$

In Equation (3), the left side refers to the change in momentum over time and the amount of momentum flowing out in the vertical direction of the control volume and the right side refers to the surface force representing the pressure acting on the control volume and the viscosity acting on the control volume.

#### 2.2.2. Calculation of Turbulent Flow

A turbulence transport model is primarily used for calculating flow.

(1)Turbulence transport model

In the analysis of indoor airflow, detailed variations in turbulent flow over time are excluded; hence, it is possible to use an averaged governing equation for a timescale larger than the timescale for turbulent flow. The time-averaged Navier–Stokes equation includes Reynolds stress ($-\rho {u\prime }_{i}u\prime_{j}$), which involves the Reynolds term along with a turbulent diffusion term ($-\rho u\prime_{i}\zeta \prime$). The turbulence model can be expressed as Equations (4) and (5) using Boussinesq’s assumption that the Reynolds stress and turbulent diffusion term are proportional to the average velocity gradient.
(4)$$\tau_{ij}=-\rho u_{i}^{\prime }u_{j}^{\prime }=u_{t}\left(\frac{\partial U_{i}}{\partial x_{j}}+\frac{\partial U_{j}}{\partial x_{i}}\right)-\frac{2}{3}\delta_{ji}\rho k$$
(5)$$-\rho u_{i}^{\prime }\zeta =\Gamma_{t}\frac{\partial Z}{\partial x_{i}}$$
where $u_{t}$ represents turbulent viscosity, $\Gamma_{t}$ represents the turbulent diffusion coefficient, $\zeta_{ij}$ is the Kronecker δ, and k represents turbulent kinetic energy. The last term of Equation (4) can be ignored because its value is very small [18]. In general, the turbulence model satisfying the Reynolds stress transport equation is a turbulent viscosity model based on Boussinesq’s assumption. Most of the models used in engineering fields in recent years are of this type. The turbulent viscosity model can be subdivided into a mixing length model, one-equation model, and two-equation model. The most commonly used model for predicting the distribution of airflow and temperature in buildings is the k−ε subset of the two-equation model. The turbulent viscosity ($u_{t}$) of the general k−ε model is presented in Equation (6).
(6)$$u_{t}=C_{D}\rho \frac{k^{2}}{\varepsilon }$$
where ε represents the dissipation rate of turbulent kinetic energy and $C_{D}$ is an empirical constant with a value of 0.09 in the present case. Because the values of k and ε are required to calculate the turbulent viscosity, the two equations are added [29].

The values of k and ε can be obtained from the following turbulent transport equations: Equations (7) and (8).
(7)$$\frac{\partial (\rho k)}{\partial t}=\frac{\partial (\rho ku_{i})}{\partial x_{i}}=\frac{\partial }{\partial x_{j}}\left[\frac{\mu_{t}}{\sigma_{k}}\frac{\partial k}{\partial x_{j}}\right]+2\mu_{t}S_{ij}S_{ij}-\rho \varepsilon$$
(8)$$\frac{\partial (\rho \varepsilon )}{\partial t}=\frac{\partial (\rho \varepsilon u_{i})}{\partial x_{i}}=\frac{\partial }{\partial x_{j}}\left[\frac{\mu_{t}}{\sigma_{\varepsilon }}\frac{\partial \varepsilon }{\partial x_{j}}\right]+C_{1\varepsilon }\frac{\varepsilon }{k}2\mu_{t}S_{ij}S_{ij}-C_{2\varepsilon }\rho \frac{\varepsilon^{2}}{k}$$

The turbulence model constants $S_{ij}=\left(\frac{\partial u_{i}}{\partial x_{j}}+\frac{\partial u_{j}}{\partial x_{i}}\right)$ and k−ε used in the analysis are as follows:$$\begin{array}{l} C_{\varepsilon 1}=1.44\\ C_{\varepsilon 2}=1.92\\ C_{D}=0.09\\ \sigma_{\varepsilon }=1.30\\ \sigma_{k}=1.00 \end{array}$$

## 3. Analysis Model

In this section, we propose an optimization method for the design variables regarding the shape of the outlet of the air circulation system through a Taguchi table of orthogonal arrays. The shape of the air circulation system was modeled using the CATIA design program, and the flow analysis was performed using STAR-CCM+. MINITAB was used as the program for the Design of Experiment [30,31,32].

### 3.1. Shape Model

To circulate the air at the top and bottom of the room, a method of suctioning the air from the bottom and discharging it to the top was selected, and a 200 W sirocco fan was utilized to force air circulation. The sirocco fan is a centrifugal multiblade-type blower that is different from general fans and has very low noise and vibration compared to its capacity. To minimize the dead zone, the air flow path was designed to have a circular column shape with a diameter of 290 mm, and the outlet height was approximately 2300 mm. Moreover, the model was simplified into a shape in which the sirocco fan was removed to perform the flow analysis. Modeling was performed as shown in Figure 2.

### 3.2. Application of the Taguchi Method

The outlet shape of the air circulation system was composed of four factors at three levels each based on a table of orthogonal arrays, which is a characteristic feature of the Taguchi method. This table of orthogonal arrays can reduce the number of experiments by confounding the secondary and higher-order interactions that have no technical significance on the main effects, such as evaluating stability and determining optimal conditions. When there are many factors, too many experiments can be performed with a general factorial design. Thus, the Taguchi method is widely used to obtain a large amount of data with a small number of experiments [33,34].

In this study, considering the multivariate layout of the design factors used, 81 fluid analyses should be performed; however, only nine flow analyses need to be performed according to the L9 orthogonal array table.

Figure 3 and Table 1 present the design variables for optimizing the air circulation system. The outlet shape model using the L9 orthogonal array table is shown in Figure 4, and the analysis was conducted on the nine models according to the outlet shape.

Table 2 lists the characteristic value design variables for the outlet shape of the air circulation device, obtained using the orthogonal array table.

The flow structure of all the models has one inlet/outlet, as shown in Figure 5a, and the volume of the flow area is shown in Figure 5b, thus demonstrating the width (6 m), length (14 m), and height (2.5 m).

### 3.3. Analysis Grid and Conditions

(1)Generation of the Analysis Grid

A grid for calculating the indoor airflow was created using the mesher in STAR-CCM+. Table 3 presents the calculation grid settings.

(2)Analysis Conditions and Results

For the internal flow, the standard k−ε turbulence model was used to calculate turbulence based on a turbulent flow in a steady state. Because the maximum wind speed of the sirocco fan used in the air circulation system was 6 m/s, the air flow rate of the outlet was set to 6 m/s and the inlet was set to atmospheric pressure to perform the analysis. Moreover, 54 points were set, as shown in Figure 6, to derive the results for the airflow rate. The upper part is denoted T, the central part is denoted M, and the lower part is denoted B. When viewed from the long side, the left line is L, the middle line is C, and the right line is R, and each point can be distinguished.

Flow analysis was performed to identify the flow velocity distribution of indoor air, and Figure 7 presents the flow velocity of air in cross-sections at 1 m intervals for nine models in the horizontal and vertical directions.

In terms of the flow distribution, the majority of Models 3, 4, and 8 demonstrated a flow rate of 0 m/s, as can be seen from the cross sections. These models have the largest flow radii, and the flow rate tends to decrease with increasing flow radius. The results of the flow analysis for the airflow rates of the upper, central, and lower parts of each model are shown in Table 4, Table 5 and Table 6.

Based on the results from calculating the average flow rates of the upper, central, and lower parts derived through the flow analysis, in the upper part, Model 1 had the fastest average flow rate at 0.26 m/s and Model 3 had the slowest flow rate at 0.18 m/s. In the central part, Model 1 had the fastest average flow rate at 0.29 m/s and Model 5 showed the slowest flow rate at 0.18 m/s. In the lower part, Model 1 had the fastest flow rate at 0.26 m/s and Model 3 had the slowest flow rate at 0.15 m/s.

Based on the overall results, Model 1 appears to have the fastest flow rate and Model 3 has the slowest flow rate.

Statistical analyses were performed based on the results to ensure the reliability and optimize the process. To analyze the effect of the characteristic values obtained using the orthogonal array table presented in Table 2 on the number of levels of the design variable, an average analysis was conducted by dividing it into upper, central, and lower parts. For the average analysis, the optimal level for the characteristics was estimated as a combination of design variables of the average value for each level of each factor, and the results of each average analysis of the characteristics for the upper, central, and lower parts are shown in Table 7, Table 8 and Table 9, respectively. Figure 8 presents the average analysis results of the design variables based on the characteristic values.

Based on the average analysis results obtained through the design of experiment Taguchi method, the design variables with the largest contribution to the upper air flow rate were ordered as blade number > flow radius > output height > blade angle. Those with the largest contribution to the central air flow rate were ordered as flow radius > blade number > output height > blade angle, and those with the largest contribution to the lower air flow rate were ordered as blade number > output height > blade angle > flow radius. To maximize the effect of the air circulation device, the number of blades should be reduced or the angle of the flow radius should be small. Because of the structural difficulty of reducing the number of blades, four blades were selected as the minimum number.

## 4. Experimental Apparatus and Methods

### 4.1. Experimental Apparatus

Based on the results obtained through the analysis and design of experiment to identify the upper and lower temperature distributions through indoor air circulation in this study, a prototype for Model 1 was manufactured considering the optimized design variables: blade angle of 30°, blade number of 4, output height of 60 mm, and flow radius of 200 mm (Figure 9). An 80 W sirocco fan was used for forced circulation of air, and a T-type thermocouple was manufactured to measure the temperature at 54 indoor points, as shown in Figure 10.

### 4.2. Experimental Methods

Yokogawa DA100 and DS600 were used to measure and store the temperature data, which were collected and sent to a PC.

To identify the difference between the model without an outlet shape and the model with an optimized outlet shape, the time required for the temperature deviation in the air circulation system to decrease at wind speeds of 2 m/s and 6 m/s was measured in both models.

For the initial 30 min, only the ceiling heater was operated to force a temperature difference between the upper and lower parts, and, after 30 min, the air circulation system was operated to collect data.

The temperature was measured for 1 h, and the temperature data were stored twice per second.

## 5. Results and Discussion

During air circulation system operation, the temperature convergence condition was based on when the temperature difference between the upper and lower parts was less than 2 °C. The results were derived from the point at which the device was operated for 25 min after the heater operation.

### 5.1. Air Circulation by Natural Convection

Figure 11 shows the temperature distribution over time for natural convection. The average maximum temperature was approximately 35 °C, and even when the heater was turned off at approximately 1500 s, the temperature in the lower part did not change for approximately 450 s. Moreover, the minimum temperature deviation was approximately 2.2 °C, indicating that the temperature difference between the upper and lower parts did not converge.

### 5.2. Model without an Outlet Shape

The experimental results for the design model without an outlet shape are shown in Figure 12 and Figure 13.

Figure 12 shows the results obtained when operating the air circulation system at an air flow rate of 2 m/s. As shown in the figure, the average maximum temperature was approximately 33.5 °C, the maximum temperature deviation was approximately 12.5 °C, and the minimum temperature deviation was approximately 1.1 °C. Furthermore, it took approximately 1530 s to reach a temperature deviation of less than 2 °C after operating the air circulation system. After reaching the steady state, the temperature difference between the upper and lower parts was in the range of 1.1 °C to 1.4 °C, confirming that air circulated faster with the system than with natural convection.

Figure 13 shows the results obtained when the air circulation system was operated at an air flow rate of 6 m/s. The average maximum temperature was approximately 33 °C, the maximum temperature deviation was approximately 12.6 °C, and the minimum temperature deviation was approximately 0.8 °C. Moreover, the time required to reach a temperature deviation of less than 2 °C after turning on the air circulation system was approximately 530 s, and the temperature difference between the upper and lower parts after reaching the steady state was in the range of 0.8 °C to 1.2 °C.

In the experiments on the air circulation system without an outlet shape, when the steady state was reached, the temperature deviation tended to decrease to less than 2 °C and the temperature difference was the smallest at a flow rate of 6 m/s. A short time is required for the temperature to converge, although the airflow after the air collides with the ceiling is expected to occur in the bottom direction; therefore, some corners or top air circulation does not occur.

### 5.3. Model with an Outlet Shape

Figure 14 and Figure 15 show the experimental results for the design model with an outlet shape.

Figure 14 presents the results when operating the air circulation system at an air flow rate of 2 m/s. As can be seen in the figure, the average maximum temperature was approximately 33 °C, the maximum temperature deviation was approximately 12.5 °C, and the minimum temperature deviation was approximately 0.7 °C. Moreover, the time required to reach a temperature deviation of less than 2 °C after turning on the air circulation system was approximately 710 s, and after reaching the steady state, the temperature difference between the upper and lower parts was found to be in the range of 0.7 °C to 1.0 °C. This result confirms that the temperature deviation decreased more quickly compared with the case without an outlet shape.

Figure 15 shows the results when the air circulation system is operated at an airflow rate of 6 m/s. As shown in the figure, the average maximum temperature was approximately 33 °C, the maximum temperature deviation was approximately 13.1 °C, and the minimum temperature deviation was approximately 0.6 °C. Furthermore, the time required to reach a temperature deviation of less than 2 °C after turning on the air circulation system was approximately 440 s, and the temperature difference between the upper and lower parts after reaching the steady state was in the range of 0.6 °C to 0.8 °C. Among all experiments, this configuration obtained the best results and achieved the smallest temperature deviation. The findings indicate that the shape of the outlet of the air circulation device led to good air circulation throughout the indoor space and a short time until temperature convergence.

## 6. Conclusions

In this study, the Taguchi orthogonal array method and computational fluid analysis were used to verify the indoor air flow and enable smooth air circulation throughout a room. Based on the obtained results, a prototype was manufactured to identify the temperature distribution over time. The following conclusions were drawn.

(1) Although the temperature difference between the upper and lower parts decreased with only natural convection, the temperature difference between the upper and lower parts could not converge to less than 2 °C, despite the experiments being performed outside of winter months. This finding indicates that there may be a large temperature difference in winter. Therefore, forced circulation of indoor air is required to reduce the temperature difference between the upper and lower parts.

(2) Based on the shape optimization results, the time required to reach a temperature deviation of less than 2 °C was improved by up to 115% compared with the model without an outlet shape, and the temperature deviation was approximately 30% better after reaching the steady state. Therefore, it is believed that the shape of the outlet can send air farther away, thus contributing significantly to the indoor flow.

(3) When there was no outlet shape, air from the air circulation system hit the ceiling and air flowed directly to the floor, thus reducing the temperature deviation around the air circulation system but not the overall temperature deviation indoors. Therefore, it is expected that the air circulation was not smooth.

(4) It was confirmed that the time required to reduce the temperature difference between the upper and lower parts can be decreased from 820 s to 90 s by optimizing the shape of the outlet. As the temperature deviation remains below 1 °C, it is expected that this optimization will contribute to plant growth by uniformly maintaining the temperature of the upper and lower parts of the room in winter.

(5) The temperature distribution of the indoor air confirmed that the forced convection performance of the outlet shape of the air circulation system had changed. As shown in the analysis results, the system can be applied to various indoor spaces where temperature deviation occurs in the upper and lower parts.

(6) As suggested in this study, if a structure with an outlet shape in an air circulation system is applied to an air circulation system in an office or smart farm, then the costs of heating and cooling in winter and summer can be reduced.

It is expected that more advanced air circulation systems will be developed in the future if a speed controller for the fan is designed according to the temperature and applied to actual smart farms, which would allow experiments to be performed to investigate the effects on the growth rate of plants.

## Figures and Tables

**Figure 1 sensors-23-02570-f001:**
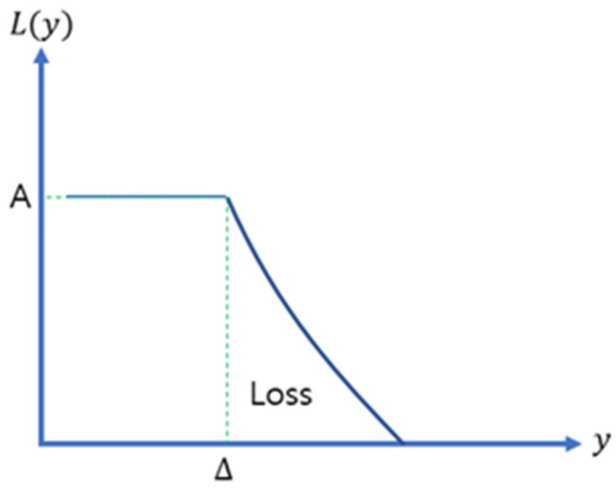
Larger-the-better characteristics.

**Figure 2 sensors-23-02570-f002:**
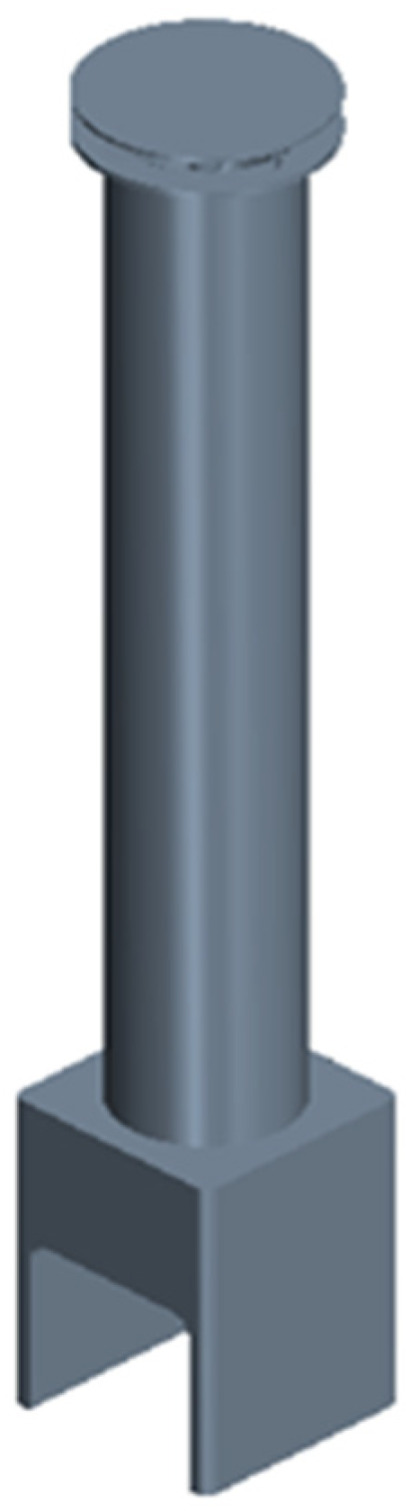
Air circulation system model.

**Figure 3 sensors-23-02570-f003:**
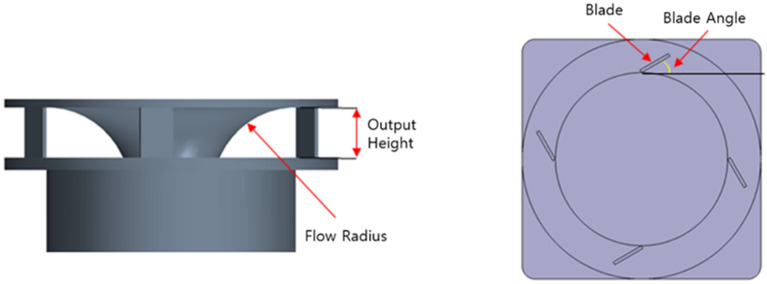
Outlet design parameters.

**Figure 4 sensors-23-02570-f004:**
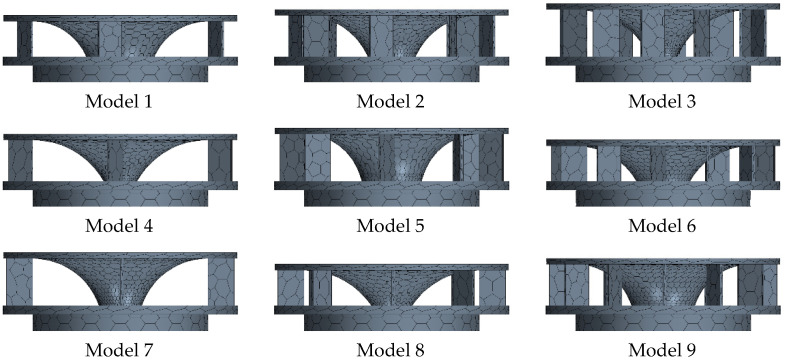
Outlet model with L9 applied.

**Figure 5 sensors-23-02570-f005:**
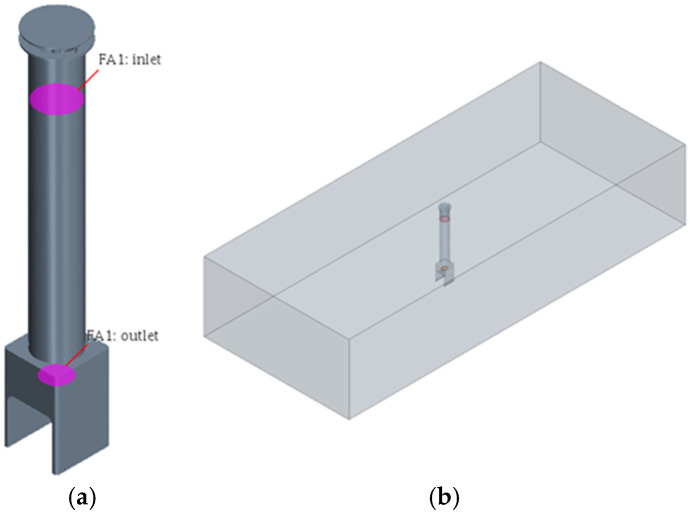
Analysis conditions and model. (**a**) Setup of the inlet and outlet; (**b**) FE model.

**Figure 6 sensors-23-02570-f006:**
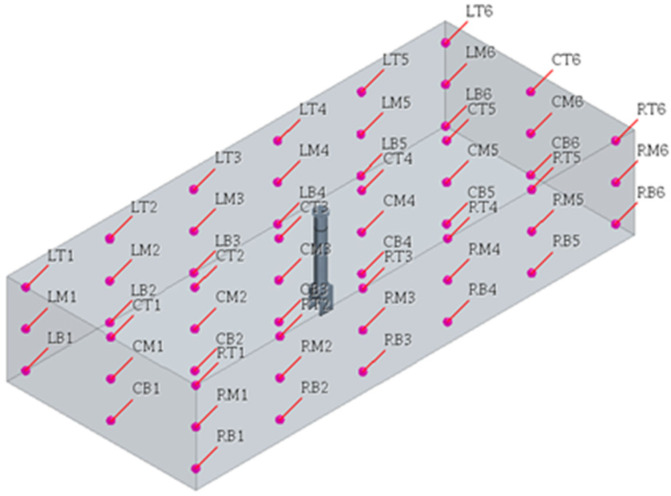
Air flow velocity points.

**Figure 7 sensors-23-02570-f007:**
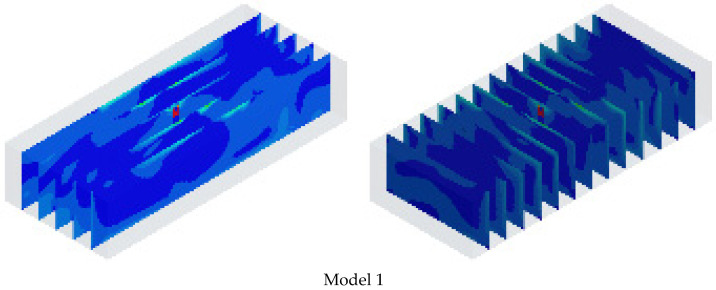
Flow analysis results.

**Figure 8 sensors-23-02570-f008:**
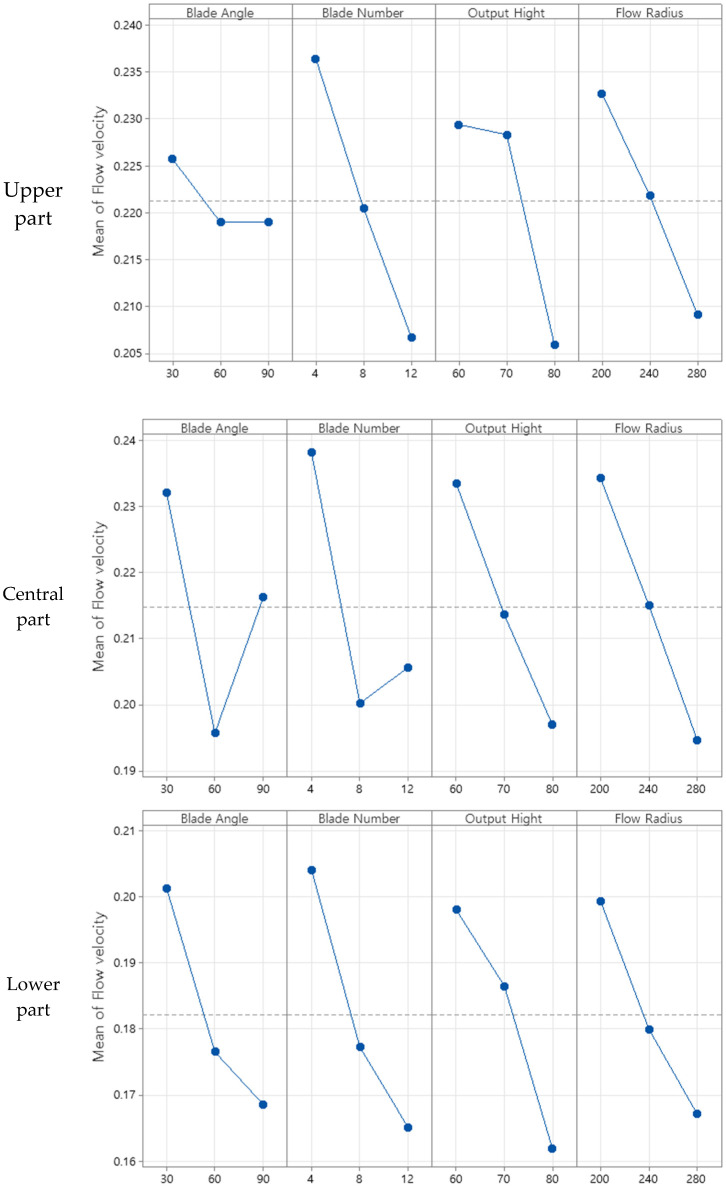
Main effects plot.

**Figure 9 sensors-23-02570-f009:**
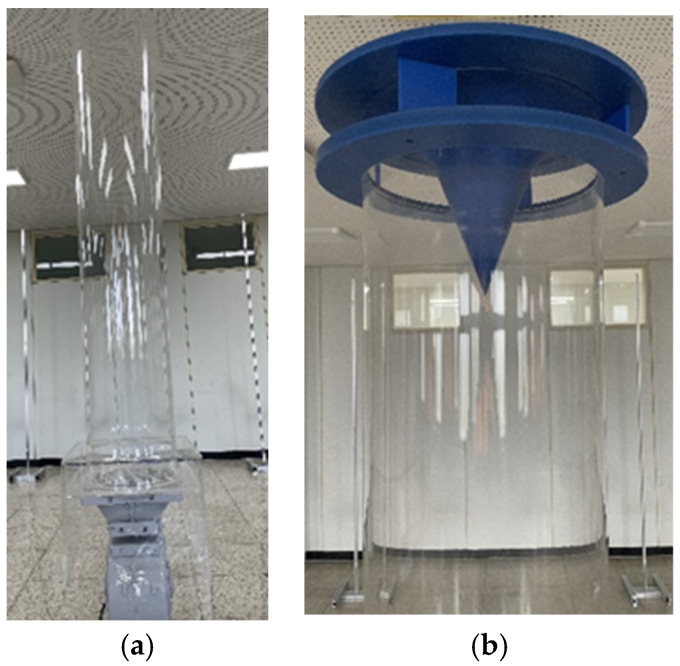
Air circulation system prototype. (**a**) Circular outlet shape; (**b**) Optimized outlet shape.

**Figure 10 sensors-23-02570-f010:**
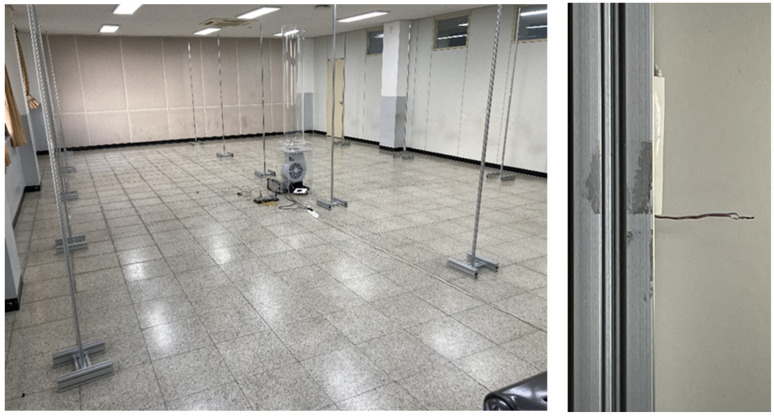
Fifty-four temperature points and air circulation system.

**Figure 11 sensors-23-02570-f011:**
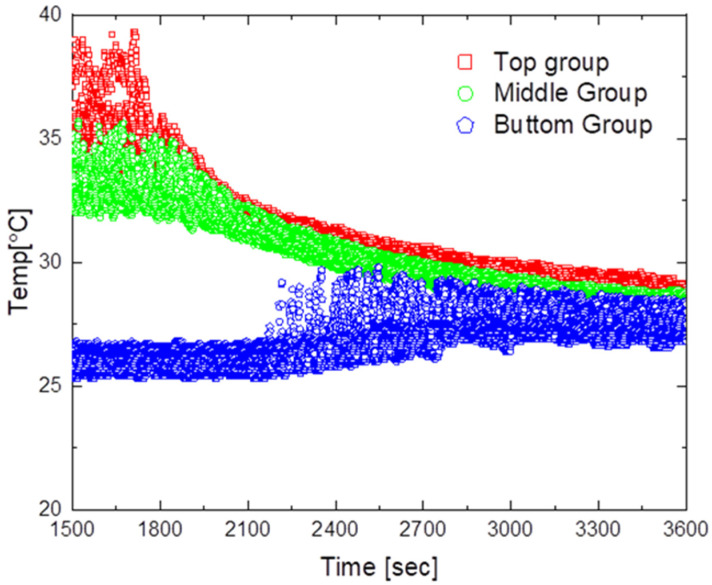
Temperature distribution by natural convection.

**Figure 12 sensors-23-02570-f012:**
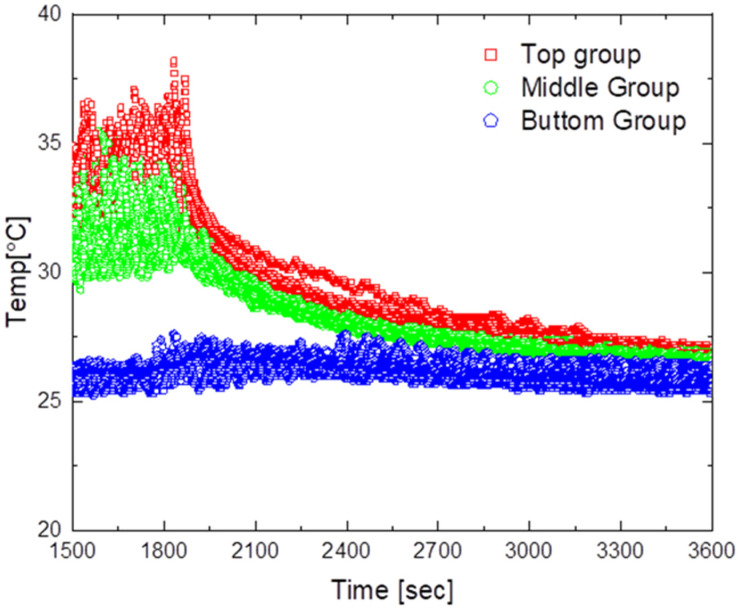
Temperature distribution for the air circulator operating at 2 m/s.

**Figure 13 sensors-23-02570-f013:**
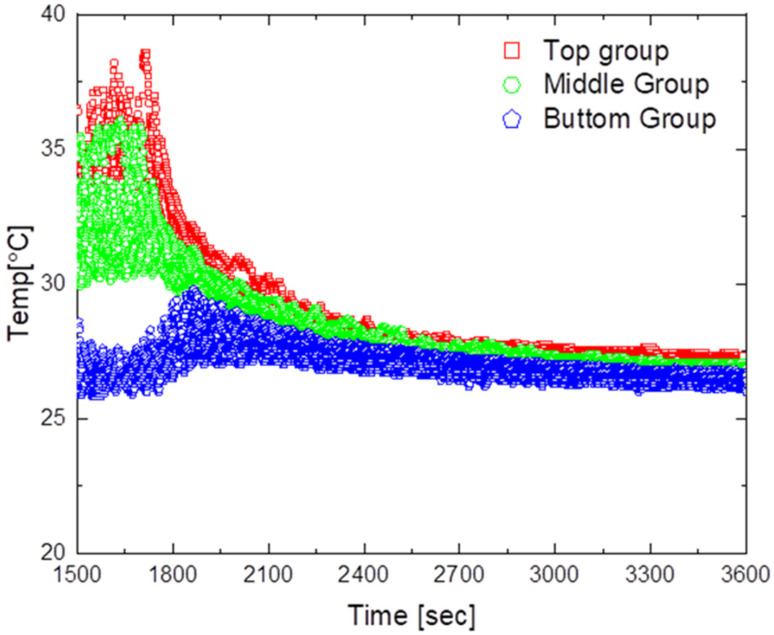
Temperature distribution for the air circulator operating at 6 m/s.

**Figure 14 sensors-23-02570-f014:**
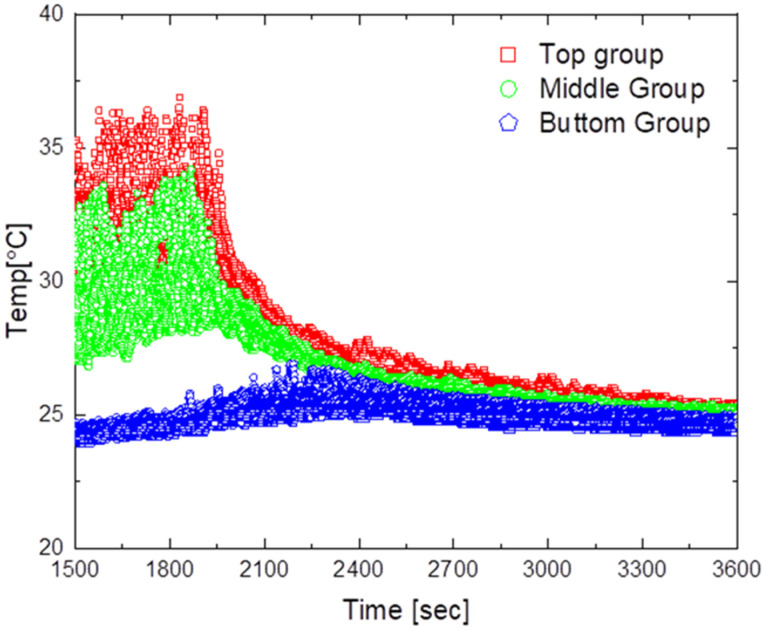
Temperature distribution for the optimized air circulator operating at 2 m/s.

**Figure 15 sensors-23-02570-f015:**
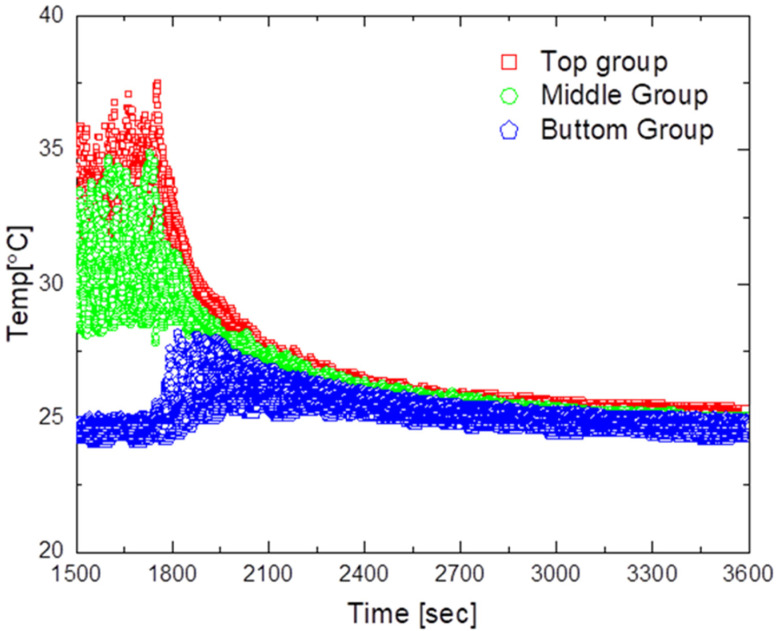
Temperature distribution for the optimized air circulator operating at 6 m/s.

**Table 1 sensors-23-02570-t001:** Levels and design variables.

	Factor	Blade Angle [°]	Blade Number [ea]	Output Height [mm]	Flow Radius [mm]
Level					
1		30	4	60	200
2		60	8	70	240
3		90	12	80	280

**Table 2 sensors-23-02570-t002:** Orthogonal array table with design variables applied.

Exp.	Blade Angle [°]	Blade Number [ea]	Output Height [mm]	Flow Radius [mm]
1	30	4	60	200
2	30	8	70	240
3	30	12	80	280
4	60	4	70	280
5	60	8	80	200
6	60	12	60	240
7	90	4	80	240
8	90	8	60	280
9	90	12	70	200

**Table 3 sensors-23-02570-t003:** Calculation grid settings used for fluid analysis.

Part	Properties	Value
Fluid Region	Mesh Type	polyhedral
	Base Size [mm]	20
	Number of Prism Layers [-]	3
	Prism Layer Thickness [mm]	0.25
	Minimum Surface Size [mm]	0.5

**Table 4 sensors-23-02570-t004:** Air flow velocity at the upper 18 points.

Units: m/s									
	Mod.1	Mod.2	Mod.3	Mod.4	Mod.5	Mod.6	Mod.7	Mod.8	Mod.9
LT1	0.228	0.280	0.224	0.344	0.121	0.069	0.144	0.120	0.161
LT2	0.113	0.777	0.340	0.368	0.323	0.226	0.244	0.333	0.268
LT3	0.333	0.094	0.204	0.306	0.355	0.232	0.334	0.373	0.350
LT4	0.255	0.210	0.355	0.352	0.351	0.372	0.341	0.241	0.343
LT5	0.570	0.143	0.183	0.261	0.195	0.281	0.360	0.407	0.299
LT6	0.206	0.192	0.057	0.125	0.163	0.131	0.128	0.149	0.269
CT1	0.125	0.258	0.017	0.199	0.144	0.153	0.199	0.107	0.145
CT2	0.162	0.386	0.097	0.079	0.120	0.165	0.126	0.103	0.083
CT3	0.187	0.137	0.136	0.112	0.141	0.174	0.134	0.085	0.134
CT4	0.167	0.131	0.181	0.125	0.142	0.125	0.138	0.094	0.138
CT5	0.156	0.057	0.194	0.092	0.123	0.095	0.071	0.133	0.090
CT6	0.208	0.141	0.084	0.071	0.155	0.170	0.167	0.123	0.176
RT1	0.263	0.033	0.094	0.138	0.186	0.151	0.151	0.156	0.124
RT2	0.551	0.213	0.230	0.261	0.263	0.317	0.346	0.359	0.327
RT3	0.239	0.189	0.198	0.392	0.387	0.369	0.311	0.279	0.302
RT4	0.598	0.106	0.315	0.234	0.332	0.272	0.332	0.262	0.325
RT5	0.131	0.670	0.312	0.468	0.253	0.363	0.265	0.396	0.350
RT6	0.197	0.171	0.091	0.199	0.107	0.175	0.161	0.139	0.132
mean	0.26	0.23	0.18	0.23	0.21	0.21	0.22	0.21	0.22

**Table 5 sensors-23-02570-t005:** Air flow velocity at the central 18 points.

Units: m/s									
	Mod.1	Mod.2	Mod.3	Mod.4	Mod.5	Mod.6	Mod.7	Mod.8	Mod.9
LM1	0.263	0.296	0.250	0.303	0.129	0.144	0.178	0.174	0.142
LM2	0.339	0.315	0.221	0.179	0.128	0.221	0.243	0.314	0.234
LM3	0.305	0.262	0.180	0.163	0.257	0.234	0.257	0.163	0.230
LM4	0.267	0.254	0.250	0.228	0.213	0.195	0.181	0.141	0.205
LM5	0.383	0.274	0.130	0.291	0.220	0.271	0.290	0.291	0.289
LM6	0.325	0.216	0.120	0.217	0.200	0.171	0.199	0.195	0.225
CM1	0.287	0.205	0.089	0.177	0.172	0.188	0.254	0.205	0.277
CM2	0.301	0.198	0.175	0.094	0.147	0.194	0.191	0.183	0.196
CM3	0.322	0.157	0.233	0.237	0.254	0.087	0.225	0.116	0.189
CM4	0.283	0.184	0.254	0.193	0.120	0.165	0.200	0.123	0.227
CM5	0.269	0.089	0.182	0.069	0.199	0.159	0.170	0.179	0.197
CM6	0.266	0.093	0.136	0.175	0.088	0.255	0.185	0.278	0.273
RM1	0.335	0.057	0.145	0.186	0.222	0.181	0.186	0.181	0.201
RM2	0.352	0.213	0.124	0.175	0.179	0.242	0.282	0.268	0.263
RM3	0.157	0.266	0.272	0.198	0.237	0.225	0.228	0.108	0.235
RM4	0.349	0.216	0.245	0.126	0.291	0.245	0.278	0.092	0.217
RM5	0.291	0.308	0.178	0.299	0.041	0.311	0.277	0.369	0.297
RM6	0.194	0.302	0.154	0.257	0.202	0.218	0.179	0.231	0.167
mean	0.29	0.21	0.19	0.21	0.20	0.23	0.25	0.23	0.27

**Table 6 sensors-23-02570-t006:** Air flow velocity at the lower 18 points.

Units: m/s									
	Mod.1	Mod.2	Mod.3	Mod.4	Mod.5	Mod.6	Mod.7	Mod.8	Mod.9
LB1	0.164	0.157	0.126	0.154	0.065	0.075	0.110	0.145	0.101
LB2	0.399	0.259	0.163	0.157	0.075	0.147	0.180	0.259	0.161
LB3	0.271	0.272	0.192	0.185	0.201	0.179	0.212	0.162	0.190
LB4	0.177	0.237	0.205	0.199	0.238	0.222	0.189	0.124	0.189
LB5	0.268	0.129	0.138	0.267	0.245	0.228	0.215	0.184	0.173
LB6	0.240	0.103	0.070	0.066	0.126	0.136	0.171	0.181	0.183
CB1	0.382	0.148	0.202	0.305	0.265	0.214	0.184	0.098	0.234
CB2	0.159	0.215	0.173	0.185	0.215	0.168	0.133	0.130	0.095
CB3	0.292	0.126	0.144	0.127	0.122	0.251	0.115	0.189	0.193
CB4	0.176	0.154	0.132	0.173	0.129	0.210	0.152	0.133	0.230
CB5	0.180	0.290	0.179	0.192	0.212	0.115	0.131	0.111	0.142
CB6	0.334	0.276	0.146	0.349	0.156	0.038	0.088	0.216	0.102
RB1	0.226	0.036	0.081	0.207	0.128	0.111	0.154	0.172	0.169
RB2	0.272	0.174	0.036	0.142	0.209	0.196	0.196	0.170	0.175
RB3	0.169	0.174	0.210	0.167	0.252	0.232	0.192	0.104	0.188
RB4	0.287	0.261	0.222	0.123	0.213	0.203	0.215	0.125	0.220
RB5	0.385	0.361	0.202	0.206	0.093	0.186	0.208	0.282	0.212
RB6	0.230	0.203	0.063	0.177	0.093	0.207	0.183	0.178	0.159
mean	0.26	0.20	0.15	0.19	0.17	0.17	0.17	0.16	0.17

**Table 7 sensors-23-02570-t007:** Upper flow rate analysis of means.

Level	Blade Angle	Blade Number	Output Height	Flow Radius
1	0.2257	0.2364	0.2294	0.2327
2	0.2190	0.2205	0.2283	0.2219
3	0.2190	0.2068	0.2060	0.2092
Delta	0.0067	0.0296	0.0234	0.0235
Rank	4	1	3	2

**Table 8 sensors-23-02570-t008:** Central flow analysis of means.

Level	Blade Angle	Blade Number	Output Height	Flow Radius
1	0.2321	0.2381	0.2334	0.2343
2	0.1958	0.2003	0.2136	0.2151
3	0.2163	0.2057	0.1970	0.1947
Delta	0.0363	0.0378	0.0364	0.0395
Rank	4	2	3	1

**Table 9 sensors-23-02570-t009:** Lower flow rate analysis of means.

Level	Blade Angle	Blade Number	Output Height	Flow Radius
1	0.2013	0.2041	0.1980	0.1993
2	0.1766	0.1773	0.1865	0.1800
3	0.1686	0.1651	0.1620	0.1672
Delta	0.0326	0.0389	0.0360	0.0321
Rank	3	1	2	4

## Data Availability

Not applicable.

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
