# Peer review of "Optimization of the Outlet Shape of an Air Circulation System for Reduction of Indoor Temperature Difference"

_sensors, 2023, doi:10.3390/s23052570_

Round 1
Reviewer 1 Report
The materials and methods still not clear: what, who, where, when and how,
especially in "what" and "how" author carry on the study and analysis.
The results and discussion still very shallow, and additional data and graphical things recommended.
Reviewer 2 Report
This paper is two fold. First a numerical analysis is done on flow in box with a fan inside. Nine simulations are done with different types of outlet of the fan (variations in blade numbers, angle, flow radius and outlet height). Using this the optimal model outlet was chosen. Then experiments were conducted in a real sized room (1) natural convection, 2) fan but not optimal fan outlet, 3) fan using the optimal fan outlet). Natural convection leads to unsatisfactory results, fan but not optimal fan outlet is better but fan with optimal fan outlet is best. There is no comparison of the numerical results and the experiments which I find odd. The numerical results are only used to find the optimal outlet but if the simulations are not leading to results that can be verified by the experiments then it is hard to know if they have actually lead to the optimal outlet. So this needs to be addressed. One other thing I find odd. You clearly have temperature differences but in the theoretical background you do not specify the equation for conservation of energy which is essential to catch the temperature variation. The methodology needs to be further explained, especially the one on the numerical method and choices made there (more detailed explanation below). The literature is current but is almost exclusively (possibly one exception) based on work by Korean researchers. This is not a subject that is exclusively relevant to South Korea so there must some research by others nationalities that is worth mentioning. This makes me (and I'm sure others too) very skeptical on the quality of the work and must be fixed before publication. I think the English could be improved. Sometimes the flow is not as good as it could be. For example in line 118, loss function and y have never been mentioned so it would be more natural to use "a" rather than "the" (this appear often). I think the sentence on y needs to be rephrased or the formula to come before to make this clearer. The subject is interesting even though it is not given that the results will transfer to the application suggested (smart farms). Actually it is guarantied that the results will not exactly be the same but just not clear how much they will deviate from the results presented in the paper. Despite that I think the results are interesting and can lead to useful applications in ever more sustainable future. I suggest it to be published after the comments here have been addressed. Here are some more detailed comments:Lines 51-53 " According to data released by the Ministry of Agriculture, Food and Rural Affairs, the number of smart farms increased in 2020 by 2368 units compared to that in 2017, and the area farmed increased by 1938 ha". It does not say what ministry this applies to. Is this the Ministry of Agriculture, Food and Rural Affairs in South Korea? This should be clearly stated so the reader knows what this applies to. Line 126, define what "better" means in this particular setting. Equation 1-1, there are parameters there that are not defined. Please define, A_0, Delta, and k. Equation 1-2, define variables. Rho has for example never been defined. Equation 1-5 is the well known continuity equation, why do you need to derive it? It think you can skip equations 1-2, 1-3 and 1-4 and simply have 1-5, just as you do with equation 1-6 for momentum. You mentioned earlier conservation of heat but there is no equation addressing that. If you want to catch changes in temperature this is needed. At least it needs to be addressed why you don't include it. Line 171 " The turbulence transport model is primarily used for calculating flow" there are multiple turbulent transport models (as you explain below) so use "A" instead of "The". Lines 181-182 " the last 182 term of Equation (1-7) can be ignored because its value is very small". Justify or name a reference (or both). Line 193 " which is 0.09 in the present case" justify and give a reference. Lines 193 -194 " Because a value of k and is required to calculate turbulent viscosity, two equations are added" It is not clear from the text which two you add (you specify more than two equations). So make it clear which two equations are added. Lines 203-205 "CATIA design program, and flow analysis was performed with STAR-CCM+. Furthermore, MINITAB was used as the program for the Design of Experiment". Here it would be nice to add some reference to those programs used. Where can the interested reader learn more about CATIA, STAR-CC+ and MiNITAB? If one would want to reproduce the results. Line 208-209 " Sirocco fan was utilized 208 to force air circulation" add a sentence "Sirocco fan is ...." and some explanation. Lines 209 -2011 " To minimize the dead zone, the air flow path was designed in a circular column shape with a diameter of 290 mm, and the outlet height was approximately 2300 mm " why did you choose this particular size and shape? Justify or explain. Lines 225-227 " In this study, considering the multi-variate layout of design factors used, 81 fluid analyses should be performed; however, only 9 flow analyses need to be performed according to the L9 orthogonal arrays table." How did you find those numbers? Based on what comes before it is hard to realize where those 81 analysis setups are. Some explanation is needed. Showing some intermediate results to help the reader realize how this leads to L9 orthogonal array would be helpful too. Chapter 3.3 What is the CFD number? Figure 7 the images are so small it is really hard to see what is on them Lines 285-286 " To ensure reliability and optimize the process, statistical analysis was performed based on the results". Explain this in more detail. What kind of statistical analysis? Table 6. Shouldn't this info come earlier? Figure 8. The graphs need to be enlarged and especially the numbers on the axis, the name of the y axis and the title of each graph. I also feel like the figure text needs to be more descriptive. The method surrounding ANOM needs to be further explained. Give enough explanation so the reader can replicated. Can you compare the numerical results to the experiments to see if you get similar results? If you don't get similar results how do you know if the model 1 is optimal? You show by experiment that it is better than no outlet but not that it is better than the other outlets. Did you think about the effects of radiation? Both the experiments and numerical simulations were done on an empty room but in the application suggested plants will be in the room. Those live in soil and need water so there must be some (probably multiple layers) shells holding the plants. This will hinder the flow and make convection slower. You need to address that. One cannot assume that those results will directly transfer to smart farms.
Round 2
Reviewer 2 Report
The authors have improved the paper and responded to most of the suggestions. There are some things I still believe need to be addressed:
1. One of the issues I mentioned but has not been tackled is:
"According to data released by the Ministry of Agriculture, Food and Rural Affairs" Is this the Ministry of Agriculture, Food and Rural Affairs in South Korea? Then that needs to be specified because there are similar ministries in most countries which can easily be mistaken as this one. I have never heard of a global Ministry of Agriculture, Food and Rural Affairs. I think a reference to those numbers would also be beneficial (or to a report made by those).
2. You have added some other nationalists to the reference list but I still think you must be able to add more. As we all agree on, this is a global issue.
3. The numbering of the new references is also messed up. On page 2 you have:
" CHEBOXAROV, Vik, et al. A model of a large-scale cross-flow wind turbine with rotating blades is presented and a numerical and aerodynamic study of this turbine is conducted[27]."
also
". S Murakami et al. conducted a simulation-based ventilation characteristic analysis study to apply a new scale to the indoor ventilation effect.[28]"
but in the reference list you have:
[26] CHEBOXAROV, Vik, et al. Vertical air circulation in a low-speed lateral flow wind turbine with rotary blades. Technical Physics Letters, 2008, 34.1.
[27] MURAKAMI, Shuzo; KATO, S. New scales for ventilation efficiency and their application based on numerical simulation of room airflow. In: Proceedings of International Symposium on Room Air Convection and Ventilation Effectiveness, University of Tokyo. 1992. p. 22-38.
[28] Yoon. H. J, Ham. I. T, Kim. J. S, Choi. J. D, “Optimization of the Manufacturing Process for Boiled-dried Anchovy Using Response Surface Methodology (RSM)”, Journal of Fisheries and Marine Sciences Education, Vol. 29, No. 6, pp. 1984-1993, 2017
So in the reference list 27 is not CHEBOXAROV, Vik, et al, and 28 is not Murakami. This needs to be fixed and possibly other similar mix-ups.
4. I def. think a native English speaker should read the paper and make adjustments to the text.
